# Investigation of Focusing Properties on Astigmatic Gaussian Beams in Nonlinear Medium

**DOI:** 10.3390/s22186981

**Published:** 2022-09-15

**Authors:** Shishi Tao, Jiayun Xue, Jiewei Guo, Xing Zhao, Zhi Zhang, Lie Lin, Weiwei Liu

**Affiliations:** 1Institute of Modern Optics, Eye Institute, Nankai University, Tianjin 300350, China; 2Tianjin Key Laboratory of Micro-Scale Optical Information Science and Technology, Tianjin 300350, China; 3Tianjin Key Laboratory of Optoelectronic Sensor and Sensing Network Technology, Tianjin 300350, China

**Keywords:** laser filamentation, nonlinear effects, astigmatic Gaussian beams, remote detection

## Abstract

Ultra-short laser filamentation has been intensively studied due to its unique optical properties for applications in the field of remote sensing and detection. Although significant progress has been made, the quality of the laser beam still suffers from various optical aberrations during long-range transmission. Astigmatism is a typical off-axis aberration that is often encountered in the off-axis optical systems. An effective method needs to be proposed to suppress the astigmatism of the beam during filamentation. Herein, we numerically investigated the impact of the nonlinear effects on the focusing properties of the astigmatic Gaussian beams in air and obtained similar results in the experiment. As the single pulse energy increases, the maximum on-axis intensity gradually shifted from the sagittal focus to the tangential focus and the foci moved forward simultaneously. Moreover, the astigmatism could be suppressed effectively with the enhancement of the nonlinear effects, that is, the astigmatic difference and the degree of beam distortion were both reduced. Through this approach, the acoustic intensity of the filament (located at the tangential focal point) increased by a factor of 22.8. Our work paves a solid step toward the practical applications of the astigmatism beam as the nonlinear lidar.

## 1. Introduction

When a high-power ultrafast laser pulse propagates in transparent nonlinear and dispersive media such as air, it will produce a plasma channel that is called filamentation or self-guided propagation [1,2,3,4]. The main properties of filament are characterized by a contracted diameter ~100 μm and a high-intensity ~5 × 10^13^ W/cm^2^ generating an electron plasma of density ~10^16^ cm^−3^ [5,6,7] and a supercontinuum spectrum from the ultraviolet (UV) to the near-infrared (NIR) [2]. Because of its unique features, ultra-short laser filamentation has a wide variety of applications in the remote detection field, including atmospheric remote sensing and weather control, among others [8,9,10,11]. Especially, the application for the detection and identification of pollutants in the atmosphere using a lidar (Light Detection and Ranging) has vital significance for environmental protection and human health issues [12,13,14]. At present, refractive optical systems are routinely utilized to generate filament, such as combined lens consisting of concave and convex lens. Due to the limitation of optical materials, the refractive optical systems have difficult achieving a large-diameter and a lightweight design, and the existence of chromatic aberration is not conducive to the detection of fluorescence spectra. The reflective optical systems are free of the aforementioned problems and thus have great advantages in remote sensing [15,16,17]. Compared with the coaxial systems, the off-axis systems do not have the energy loss caused by central obscuration. However, there is a severe astigmatism aberration in off-axis optical systems.

The theory of linear propagation of astigmatic beams has been well reported [18,19,20]. The astigmatism aberration is manifested by the off-axis image of an object point appearing as a line or ellipse instead of a point. The existence of an astigmatism in optical system will lead to the distortion of the beam spatial phase (i.e., the wavefront). Consequently, the quality of the laser beam will be greatly altered, not only affecting the focal shift and intensity distribution, but also changing the shape and size of the light spot. Researches on focusing properties are indispensable for the controlling and developing applications of astigmatic laser beams.

Up to now, many studies on the filamentation phenomenon and mechanism of astigmatic beams have been carried out [21,22,23,24,25]. Alonso et al. studied the process of filamentation generation in air by using the astigmatic focusing lens (f = 1 m) [23]. The results showed that astigmatic focusing are achieved within longer distances producing a higher available output energy in filamentation as well as a broader spectrum with shorter pulses. Multifilament was observed (6.0 mJ) along the vertical direction. Dergachev et al. investigated the astigmatism on plasma channels formed during filamentation by using a spherical mirror (f = 25 cm) [21]. They found that a weak astigmatism increases the length of the plasma channel while a strong astigmatism can cause splitting of the plasma channel into two channels located at sagittal and tangential foci. However, the impact of the nonlinear effects on the degree of the astigmatism of the beam has not been thoroughly studied. 

In our work, we numerically demonstrate the variation of the beam spatial distribution with the propagation over 10 m by introducing an astigmatic phase factor. We find that the astigmatic difference is reduced as the nonlinear effects increases, in other words, the beam astigmatism becomes weaker. Not only are the foci moving forward, but the intensity at the tangential focus gradually exceeds that at the sagittal focus. It is noted that there are multi-filaments formed in a vertical direction behind the tangential focus. For the proof of these phenomena, we exploit the measurement method of lateral ultrasound signal to obtain the intensities of the filament at different distances in the experiment. The results confirmed the shift of the peak intensity and the variation of the focal position. The astigmatism can be suppressed through the nonlinear effects, i.e., extremely high intensities can be obtained near the focal point, which illustrates that this approach is promising for applications in long-range detection. 

## 2. Methods and Simulations

The generation of the astigmatic Gaussian beam after the laser beam reflected by a concave mirror is shown in Figure 1. It gives the spatial profile of the astigmatic beam at different propagation distances under linear condition. To start with, the profile appears as a vertical elliptical pattern until the beam is focused to the first focus (sagittal focus). After the transition of the circle of least confusion in the middle, the beam reaches the second focus (tangential focus) and diverges into a horizontal ellipse finally. The magnitude of astigmatic difference (the distance between the sagittal and tangential focal lines) is proportional to the incident angle φ. The electric field of the beam beyond the paraxial approximation in the Cartesian coordinate system is presented as:(1)$$E\left(x,y,z\right)=E_{0}\left(x,y,0\right)\exp\left(-ik_{0}z\right).$$

Moreover, the astigmatism is usually introduced as a function of the phase that modifies the wavefront of the laser beam [26]:(2)$$E_{0}\left(x,y,0\right)=\sqrt{\frac{2P_{in}}{\pi r_{0}^{2}}}\exp\left\{-\left(x^{2}+y^{2}\right)\left[\frac{1}{r_{0}^{2}}+\frac{ik_{0}}{2f}\right]\right\}\exp\left\{-ik_{0}C_{6}\left(x^{2}-y^{2}\right)\right\},$$
where $k_{0}=2\pi n_{0}/\lambda_{0}$ is the central wave number, with $n_{0}$ being the refractive index of the air and $\lambda_{0}\;$being the central wavelength. $r_{0}$ is the initial beam waist, and f is the focal length of the concave mirror. Furthermore, $C_{6}$ is the astigmatism coefficient, which is extracted from the Zernike polynomial [26]. Finally, $P_{in}=En/t_{0}$ is the input power, where En is the single-pulse energy and $t_{0}$ is the pulse duration.

The theoretical model of laser pulses propagating in a nonlinear medium can be expressed by the following partial differential equation [27]:(3)$$2ik_{0}\frac{\partial E}{\partial z}+\Delta_{⊥}E-k_{0}k_{2}\frac{\partial^{2}E}{\partial \tau^{2}}+2\frac{k_{0}^{2}}{n_{0}}n_{nl}E=0.$$

In Equation (3), the second term corresponds to the linear diffraction of the beam, and the third term describes the group velocity dispersion (GVD) of the pulse. Because our study is mainly concerned with the spatial distribution of the laser pulses, we do not consider the group velocity dispersion. The last term represents the influence of the nonlinear refractive index $n_{nl}$, including the self-focusing caused by the optical Kerr effect and the defocusing induced by the plasma refraction. Therefore, the total refractive index consists of two nonlinear components: $n_{nl}=\Delta n_{kr}+\Delta n_{p}$. In this case, the nonlinear refractive contributed by the Kerr effect is given by $\Delta n_{kr}=n_{2}I$, where $n_{2}$ is the nonlinear refractive index of the air and I is the laser intensity. Additionally, the refractive index due to plasma generation in air is approximated as $\Delta n_{p}=N_{e}\left(I\right)/2N_{crit}$ [28]. Here the critical plasma density is written as $N_{crit}=\varepsilon_{0}m_{e}{\omega_{0}}^{2}/e^{2}$, where e and $m_{e}$ are the charge and mass of the electron and $\omega_{0}$ is the laser angular frequency. The electron density $N_{e}$ is given by ionization: $dN_{e}\left(z,t\right)/dt=R\left(I\right)\times N\left(z\right)$ with $N\left(z\right)$ being the density as a function of the propagation distance *z* and $R\left(I\right)$ being the ionization rate.

In our simulation, we calculate the spatial domain numerical solution of Equation (3) based on the Crank–Nicolson method, which is one of the finite difference methods (FDM). Each dimension of the three-dimensional spatial domain is discretized as follows:(4a)$$z_{q}=z_{0}+q\Delta z,q=0,1,\ldots \ldots ,Q$$
(4b)$$x_{m}=x_{0}+m\Delta x,m=0,1,\ldots \ldots ,M$$
(4c)$$y_{l}=y_{0}+l\Delta y,l=0,1,\ldots \ldots ,L,$$

To simplify, we take $E_{m,l}^{q}$ as $E\left(x_{m},y_{l},z_{q}\right)$. Therefore, the partial differential form in Equation (3) can be converted to the differential form:(5)$$2ik_{0}\frac{E_{m,l}^{q+1}-E_{m,l}^{q}}{\Delta z}+\frac{1}{2\Delta^{2}}\left[\left(E_{m+1,l}^{q+1}+E_{m,l+1}^{q+1}-4E_{m,l}^{q+1}+E_{m-1,l}^{q+1}+E_{m,l-1}^{q+1}\right)\right.\;\;+\left.\left(E_{m+1,l}^{q}+E_{m,l+1}^{q}-4E_{m,l}^{q}+E_{m-1,l}^{q}+E_{m,l-1}^{q}\right)\right]+2\frac{k_{0}^{2}}{n_{0}}\Delta nE_{m,l}^{q}=0,$$
where Δ=Δx=Δy. The spatial domain numerical solution of Equation (3) can be obtained by introducing the initial field into the Equation (5). We assume an incident pulse with a central wavelength of 800 nm, a pulse duration (FWHM) of 60 fs, and a beam waist (FWHM) of 1 cm. Accordingly, the nonlinear refractive index in air is $n_{2}=0.5\times 10^{-19}{\mathrm{cm}}^{2}/W$ and the astigmatic coefficient is $C_{6}=0.005$. In addition, we set the range of the input pulse energy from 0.5 mJ to 4.5 mJ, corresponding to the enhancement of the nonlinear effects.

First, the spatial profile evolutions of the astigmatic Gaussian beam focused at z=10 m under nonlinear conditions are investigated numerically. Figure 2a shows the intensity profiles in the x-z and y-z plane at different energies, respectively. The beam narrows first and then widens in the x-z plane as it propagates, while the opposite situation occurs in the y-z plane. It can be inferred that the beam converges at the sagittal focus initially and converges on the tangential focus at a later point. While the input pulse energy is 0.5 mJ, the intensity distribution is similar to the linear condition. With the increase in single pulse energy, the beam changes gradually from a single filament to multi-filaments after being focused at the tangential focus. It is a result of the strong energy interchange between the background reservoir and the parts of the beam moving toward the axis [29]. Figure 2b depicts the intensity distributions in the x-y plane at z=12 m (the white dashed lines in Figure 2a). It can be clearly seen that, as the self-focusing effect becomes stronger, the light spot begins to split and multi-filaments are formed in the vertical direction (sagittal focus direction). It is particularly noteworthy that the number of foci in the vertical direction increase rapidly with the energy enhancing from 2.5 mJ to 3.5 mJ. These are consistent with the results observed in the experiments by Alonso et al. [23]. In addition, we extracted the on-axis intensity at different energies from Figure 2a. As expected, the on-axis intensity has a maximum value near the sagittal and tangential focus, expressed as $I_{1}$ and $I_{2}$, respectively. Meanwhile, another peak intensity appears on the axis after the tangential focus denoted by $I_{3}$, as shown in Figure 2b. This phenomenon is caused by the refocusing as the intensity increases in the background reservoir.

Next, we evaluate the impact of the nonlinear effects on the focusing properties in detail. Figure 3a,b summarize the variation of magnitude and the location of the maximum on-axis intensities $I_{1}$, $I_{2}$ and $I_{3}$ at different single pulse energies in Figure 2c,d, respectively. It is shown in Figure 3a that the peak intensity $I_{1}$ generated at the sagittal focus exhibits a positive linear relationship with the pulse energy $E_{n}$, which is less dependent on the self-focusing effect. On the other hand, the trend of the peak intensity $I_{2}$ is more significant than $I_{1}$ when the pulse energy is greater than 2.5 mJ. This result may be explained by the post-compression of the astigmatic beam during filamentation [23]. The elliptical beam has a higher critical power for self-focusing than the circular beam due to its asymmetry [30]. Thus, it is possible to obtain higher intensity at the tangential focus as the contribution of the Kerr effect becomes more pronounced. With the accumulation of the energy in the background reservoir, the third peak intensity $I_{3}$ shows up and rises gradually. As the input energy increases, the positions of $I_{1}$, $I_{2}$ and $I_{3}$ all move forward (compared to the propagation direction). The variations of the diameter (FWHM) and the displacement of the sagittal and tangential focal lines at the different single pulse energies extracted from Figure 2a are shown in Figure 3c,d, respectively. As the pulse energy increases, the diameters of the sagittal and tangential focus both decrease and their positions move forward. The nonlinear contribution of the refractive index due to the Kerr effect is related to the laser intensity given by $I=2{\left|A\right|}^{2}/\eta$. For such an astigmatic Gaussian beam, its intensity decays radially from the center to the edge; thus, the beam has a converging wavefront resulting in the continuous shrinking of the diameter at the focused spots. Moreover, the nonlinear effects have a more significant impact on the tangential focus, which agree with the change of the maximum on-axis intensities in Figure 3a,b.

Figure 3e shows the variation of the astigmatic difference under the different single-pulse energies, where $L_{T}-L_{S}$ is the distance between the tangential and sagittal focal lines, and $L_{I2}-L_{I1}$ is the distance between $I_{2}$ and $I_{1}$. The degree of the astigmatism becomes weaker with an evident reduction of the astigmatic difference as the pulse energy increases. To get a deeper insight into the quality of the astigmatic Gaussian beam, the astigmatic parameter β is used to describe the degree of distortion. According to the definition of the second-order moment, the square of the beam width $w_{j}\left(j=x,y\right)$ can be obtained as follows [31]:(6)$$w_{j}^{2}=\frac{4\iint {\left(j-\bar{j}\right)}^{2}I\left(x,y,z\right)dxdy}{\iint I\left(x,y,z\right)dxdy}$$
where j=x,y and $\bar{j}$ is the center of gravity. The astigmatic parameter β is defined as the ratio of the beam width $w_{j}\left(j=x,y\right)$ in the x and y directions. While the value of β is further away from 1, the shape of the spot tends to be more elliptical, i.e., the distortion of the beam is more serious. Figure 3f illustrates the variation of the astigmatic parameter β along the propagation axis at different energies. The positions of the extreme values correspond to the sagittal and tangential focal lines. By increasing the input energy, the central energy in the cross-section of the beam becomes more concentrated due to the inhomogeneity of the intensity distribution. Thus, the beam widths at the focal points decrease and the symmetry of the beam becomes better as the foci move forward, which indicates that the nonlinear effect can reduce the astigmatism.

## 3. Experiments

The optical properties of astigmatic beam during filamentation are characterized by using the optical setup in Figure 4a. In our experiments, a Ti: sapphire femtosecond laser amplifier (Legend, Coherent Inc., Santa Clara, CA, USA) is employed, which delivers 60 fs pulses with a central wavelength of 800 nm and a repetition rate of 500 Hz. The beam waist (FWHM) is about 4 mm and the maximum single pulse energy is about 3.6 mJ. The pulse energy is adjusted by using an energy attenuator consisting of a half-wave plate and a polarizing cubic crystal. A concave lens and a concave mirror with the focal lengths of −400 mm and −2000 mm, respectively, are combined to generate a filament at 10 m. The laser beam radius (FWHM) on the front surface of the concave mirror is 1 cm and the incident angle is 12°. To properly represent the ultrasonic intensity at different locations, an ultrasonic probe with a bandwidth of 4.5 MHz is mounted at a position of about 1 cm from the filament and moved along the beam propagation direction [32]. The ultrasonic signal is measured ten times at each location and averaged. To study the internal pattern of the filament at high power, we use the thermal paper fixed on the chopper to record the beam profiles at different distances. In addition, the rotation frequency of the chopper is set to 140 Hz and the exposure time of the shutter is set to 2 ms to avoid forming holes in the paper.

Figure 4b gives the beam profile with a pulse energy of 3.5 mJ at the different distances recorded on the thermal paper. We can note that the burning traces related to the intensity profiles are converted from the vertical to the horizontal direction. When the chopper is moved to the distance of 10.84 m, the intensity is concentrated in the vertical direction again. This indicates that the multi-filaments formed by the refocusing are produced in the vertical direction, which is analogous to the pattern in the Figure 2b. The ultrasonic intensity as a function of the pulse energy is shown in Figure 4c. As predicted, the intensity trends obtained in the experiment match the numerical results in Figure 4e very well. However, at 3 mJ, the experimental result of the intensity $I_{2}$ is quite different from the simulation result, which may be the intensity disturbance due to nonlinear phase changes with a high degree of instability. The comparison of the amplitudes of the maximum intensities $I_{1}$ and $I_{2}$ with the simulation results can be easily noticed in Figure 4d. The maximum acoustic intensity $I_{1}$ and $I_{2}$ are increased by 3.8 times and 22.8 times, respectively. It is a reliable verification that the nonlinear effects make a greater difference to the tangential focus. 

In addition, the variation of the astigmatic difference is basically consistent with the simulation results as shown in Figure 4e. However, the astigmatic difference increases when the input energy is 3.5 mJ. The reason is presumed to be the intensity fluctuation caused by multiple filament competition.

## 4. Conclusions

In summary, we have explored the influence of the nonlinear effects on the focusing properties of astigmatic beams during filamentation from experiments and numerical simulations, respectively. The results show that the maximum on-axis intensity gradually shifts from the sagittal focus to the tangential focus and the foci move forward simultaneously as the pulse energy increases. More importantly, the nonlinear effects are able to reduce the astigmatic difference. The long-distance transmission of the astigmatic beam and its applications have always been the main emphasis of our work, especially after discovering the suppression of the astigmatism by the nonlinear effects. These findings provide a solid theoretical basis for an application study of the astigmatic beams, e.g., for atmospheric remote sensing, as well as for the detection of pollutants in the atmosphere based on nonlinear lidar. When the astigmatism is overcome, a stronger intensity can be achieved by focusing with a concave mirror. However, limited by the maximum single pulse energy of the laser, the reduction of the astigmatism has not yet reached our expectation, and some auxiliary tools are still needed to correct the astigmatism, such as free-form surface.

## Figures and Tables

**Figure 1 sensors-22-06981-f001:**
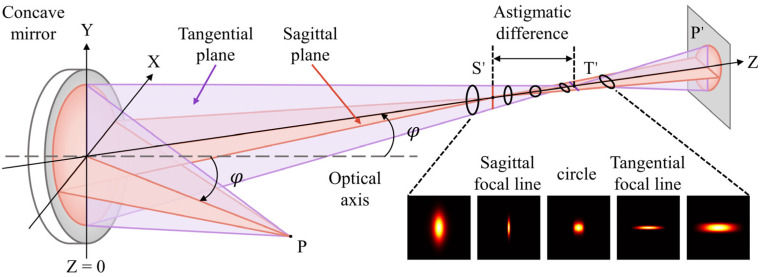
Schematic of astigmatic aberration produced by a concave mirror. Assume a femtosecond laser pulse with Gaussian spatial profile is incident to a concave mirror at angle of φ and focused along the z direction. The optical axis is the normal direction of the concave mirror and the incident angle φ is in the sagittal plane. It is worth noting that if φ is in the sagittal plane, the sagittal focal line appears first, and the tangential focal line appears after. If φ is in the tangential plane, the sign of the astigmatism is opposite.

**Figure 2 sensors-22-06981-f002:**
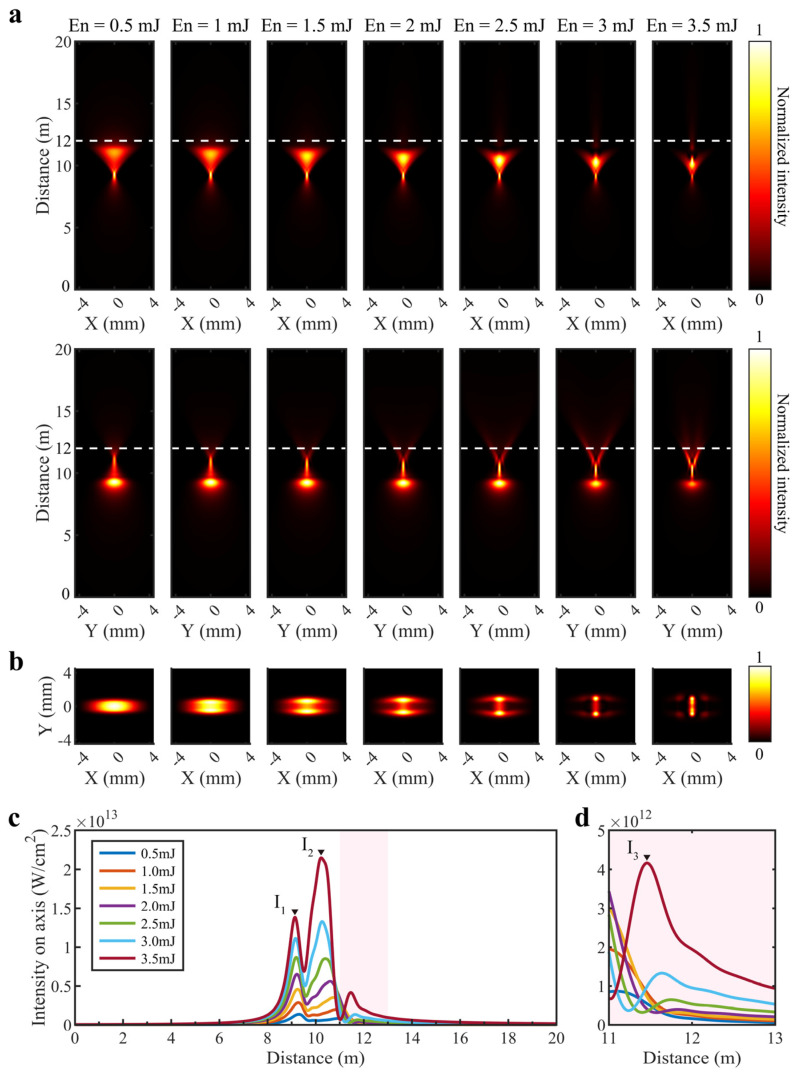
Numerical simulation results of the astigmatic Gaussian beam focused at 10 m under nonlinear conditions: (**a**) Intensity profiles in the x-z and y-z plane as a function of propagation distance at different single pulse energies; (**b**) Intensity distributions in the x-y plane at z=12 m (the white dashed lines in (**a**); (**c**) On-axis intensity extracted from (**a**); (**d**) The enlarged part of the pink area in c is shown in (**d**).

**Figure 3 sensors-22-06981-f003:**
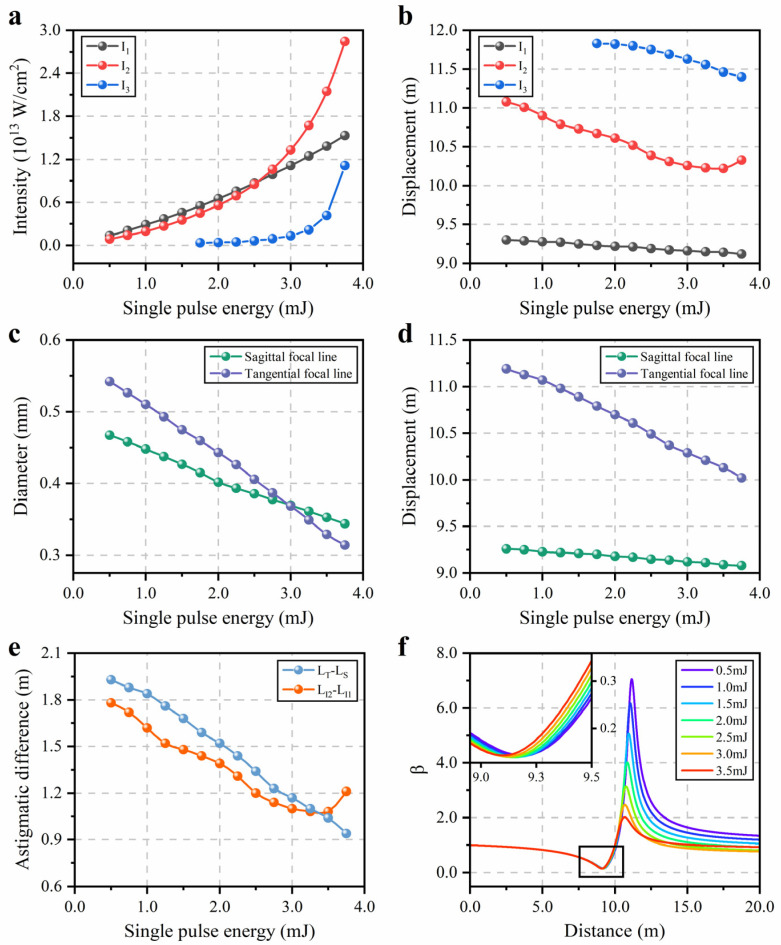
Focusing properties of the astigmatic Gaussian beams during filamentation: (**a**) Variation of the maximum on-axis intensities
$I_{1}$, $I_{2}$ and
$I_{3}$ versus single pulse energies; (**b**) Displacement of
$I_{1}$, $I_{2}$ and
$I_{3}$ versus single pulse energies; (**c**) Variation of diameter (FWHM) of the sagittal and tangential focal lines versus single pulse energies; (**d**) Displacement of the sagittal and tangential focal lines versus single pulse energies; (**e**) Astigmatic difference defined as distance between the sagittal and tangential focal lines and distance between
$I_{1}$ and
$I_{2}$ at different single pulse energies; (**f**) Variation of astigmatic parameter
β as a function of propagation distance at different single pulse energies.

**Figure 4 sensors-22-06981-f004:**
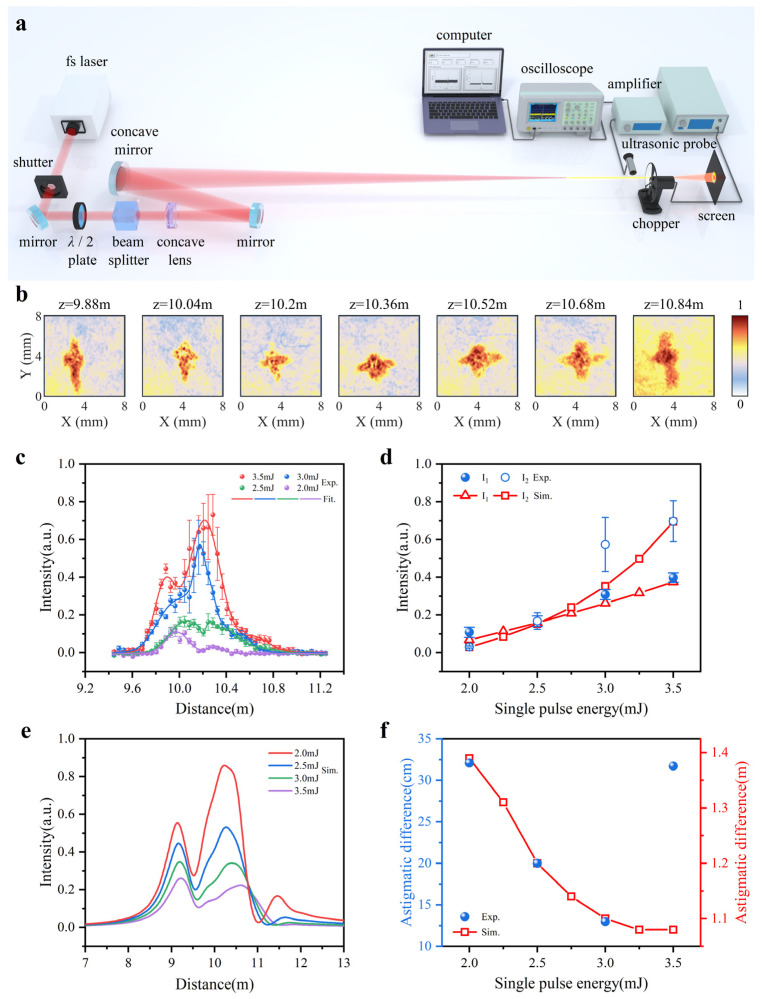
Experimental setup and results: (**a**) Experimental setup; (**b**) Thermal paper recorded laser beam intensity distribution at different distances with a pulse energy of 3.5 mJ; (**c**) Evolution of the ultrasonic intensity as a function of propagation distance at different single pulse energies; (**d**) Variation of the maximum ultrasonic intensities $I_{1}$ and $I_{2}$ versus single pulse energies compared to simulation results; (**e**) Simulation results of on-axis intensity as a function of propagation distance at different single pulse energies; (**f**) Astigmatic difference expressed as the distance between $I_{1}$ and $I_{2}$ at different single pulse energies compared to simulation results.

## Data Availability

The data presented in this study are available on request from the corresponding author.

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
