# Peer review of "Investigation of Focusing Properties on Astigmatic Gaussian Beams in Nonlinear Medium"

_sensors, 2022, doi:10.3390/s22186981_

Round 1
Reviewer 1 Report
The author experimentally studied the process of the astigmatic Gaussian beam propagating over long range in nonlinear media. They found that the enhancement of nonlinear effects reduce the astigmatism. Fluorescence with high signal-to-noise can be generated which can find application in remote detection of atmospheric pollution. This manuscript is well written, and the representation is clear and can be accepted for publication.
Reviewer 2 Report
The authors studied the influence of nonlinearity on the focusing properties of astigmatic Gaussian beams in air, and confirmed it by the results in experiment. The results obtained in this paper may be useful for applications in the field of remote sensing and detection. I recommend the manuscript for published after addressing the following comments:
1. Based on Eq. (3), the authors investigated the nonlinear effect on the focusing properties of astigmatic Gaussian beams in air. Please show how to solve Eq. (3) in detail.
2. The authors state that that the astigmatic beams propagation over long distance in nonlinear media is studied. Please explain how to define the “long propagation distance”.
3. Please give the physical explanations for the main results obtained in this paper.
Reviewer 3 Report
The authors describe simulations and experiments of nonlinear propagation of an astigmatic (initially Gaussian) beam. The simulation and experiment focus to a 10-meter distance. Although the results from both simulation and experiment are fundamentally sound, I do not see outstanding novelty in this work. A quick search yields dozens of papers which study astigmatic beams undergoing nonlinear propagation which results in filamentation: Ivanov et al. (2018) Proc. International Conf. on Atom and Molec. Pulsed Lasers XIII; Fibich et al. (2004) Opt. Lett. 29(15) 1772-1774; Dergachev et al. (2014) Quantum Electron. 44, 185; etc. Many of these papers similarly model the intensity and transverse profile of the beam as a function of distance. Curiously, none of these works are cited by the authors. Unless the authors can improve novelty, I recommend this manuscript cannot be published. Currently, the authors hardly increment on existing literature. Moreover, the manuscript should be thoroughly reviewed for spelling and grammar.
Specific comments:
Page 2, lines 62-63: It is misleading to say low average power results in less prominent nonlinear effects, especially if the reference used a tighter focusing lens such that intensity might be comparable to that achieved with longer distance measurements.
Page 2, lines 69-70: Has this ‘lateral ultrasound’ method been used previously to determine filament intensity? If so, it should be appropriately referenced.
Reviewer 4 Report
Paper is devoted to the study of the impact of nonlinear effects on the focusing properties of astigmatic Gaussian beams in air. The effect of astigmatism of Gaussian beam on the filamentation process numerically and experimenatally was shown.This work has practical applications for remote-sensing. There are some question for experimental results:
- needs to indicate bandwidth of ultrasonic probe,
- Why is the intensity dependence on the pulse energy in Fig.4d (dependence I2 Exp.) not monotonic? Namely, point at the 3 mJ is very different from the dependence obtained during modeling.
- I recommend adding a confidence interval for the dependency in Figure 4d.
- For a better perception of fig.4 c b fig.4 e , it is worth using the same dimension scale.
After all comments have been eliminated, the paper can be published.
Reviewer 5 Report
Review for the manuscript (Sensors/MDPI) 1805845
Investigation of focusing properties on astigmatic Gaussian 2 beams in nonlinear medium
Shishi Tao, Jiayun Xue, Jiewei Guo, Xing Zhao, Zhi Zhang, Lie Lin and Weiwei Liu.
The authors explored in this special issue paper, the influence of nonlinear effects on the focusing properties of astigmatic beams during filamentation, using numerical simulations, and confirmed by experiments. As an important results, they showed that the nonlinear effects are able to reduce the astigmatic difference. In addition, authors demonstrated some properties related to the pulsed laser beams as function of nonlinear properties of the propagation medium.
The paper is well written, the results and experiments are also presented in good manner.
The paper could be published in the journal Sensors (MDPI) after responding to my main objection.
The authors presented all we need as results, about the effect of non linear proprieties on the quality of the astigmatic laser beam filamentation. However, in the whole paper we don't find any physical interpretation of these results.
As an example, why the focused spots widths change with energy pulse, how the authors interprete the shifts in on axis intensities maxima.
Why the astigmatic beam becomes circular when increasing the pulse energy.
The authors should include the physical interpretation of all results in the revised version.
Reviewer 6 Report
Authors address astigmatic Gaussian-type beams propagation in a nonlinear medium. Typically, astigmatism cannot be avoided in experiment, but virtually all available theory, with very few exceptions, deals with axisymmetric modes (and with the corresponding higher-order modes). Thus I find the topic original and very relevant. For me, the most important result of the paper is that the considered nonlinearity results in suppression of nonlinear effects at long distance. The conclusion is consistent with the evidence and arguments presented. The paper is rather well written. My only recommendation is to add a few references to papers on linear theory of astigmatic beams, e.g.:
Arnaud, J.A.; Kogelnik, H. Gaussian light beams with general astigmatism, Appl. Opt. 1969, 8, 1687-1693.
Habraken, S. J. M.; Nienhuis G., Modes of a twisted optical cavity, Phys. Rev. A 2007, 75, 033819.
Kiselev, A.P.; Plachenov, A.B., Chamorro-Posada, P., Nonparaxial wave beams and packets with general astigmatism, Phys. Rev. A 2012, 85, 043835
Round 2
Reviewer 2 Report
In the revised manuscript, the authors showed how to solve Eq. (3). But I still doubt the theorical method adopted in this manuscript. It is known that, when the femtosecond filamentation in transparent media is studied, Eq. (3) must be solved simultaneously with the equation describing the evolution of the density of electrons mainly generated by photoionization (i.e., Eq. (31) in Ref. [1]). However, the equation describing the evolution of the density of electrons is not considered in this manuscript. Please explain it.
Reviewer 5 Report
Since the authors revised the manuscript accordingly, the paper could be published in its present form in the Journal Sensors.
Author Response
Thanks a lot for the referee’s comment.We have uploaded the latest revised manuscript.